# State-of-the-Art Methods and Emerging Fluid Biomarkers in the Diagnostics of Dementia—A Short Review and Diagnostic Algorithm

**DOI:** 10.3390/diagnostics11050788

**Published:** 2021-04-27

**Authors:** Eino Solje, Alberto Benussi, Emanuele Buratti, Anne M. Remes, Annakaisa Haapasalo, Barbara Borroni

**Affiliations:** 1Institute of Clinical Medicine-Neurology, University of Eastern Finland, 70211 Kuopio, Finland; eino.solje@uef.fi; 2Neuro Center, Neurology, Kuopio University Hospital, 70029 Kuopio, Finland; 3Neurology Unit, Department of Clinical and Experimental Sciences, University of Brescia, 25121 Brescia, Italy; benussialberto@gmail.com; 4International Centre for Genetic Engineering and Biotechnology, 34149 Trieste, Italy; buratti@icgeb.org; 5Research Unit of Clinical Neuroscience, Neurology, University of Oulu, 90230 Oulu, Finland; anne.remes@oulu.fi; 6Medical Research Center (MRC), Oulu University Hospital, 90220 Oulu, Finland; 7A. I. Virtanen Institute for Molecular Sciences, University of Eastern Finland, 70211 Kuopio, Finland; annakaisa.haapasalo@uef.fi

**Keywords:** dementia, biomarkers, Alzheimer’s disease, frontotemporal dementia, dementia with Lewy bodies, diagnosis

## Abstract

The most common neurodegenerative dementias include Alzheimer’s disease (AD), dementia with Lewy bodies (DLB), and frontotemporal dementia (FTD). The correct etiology-based diagnosis is pivotal for clinical management of these diseases as well as for the suitable timing and choosing the accurate disease-modifying therapies when these become available. Enzyme-linked immunosorbent assay (ELISA)-based methods, detecting altered levels of cerebrospinal fluid (CSF) Tau, phosphorylated Tau, and Aβ-42 in AD, allowed the wide use of this set of biomarkers in clinical practice. These analyses demonstrate a high diagnostic accuracy in AD but suffer from a relatively restricted usefulness due to invasiveness and lack of prognostic value. In recent years, the development of novel advanced techniques has offered new state-of-the-art opportunities in biomarker discovery. These include single molecule array technology (SIMOA), a tool for non-invasive analysis of ultra-low levels of central nervous system-derived molecules from biofluids, such as CSF or blood, and real-time quaking (RT-QuIC), developed to analyze misfolded proteins. In the present review, we describe the history of methods used in the fluid biomarker analyses of dementia, discuss specific emerging biomarkers with translational potential for clinical use, and suggest an algorithm for the use of new non-invasive blood biomarkers in clinical practice.

## 1. Introduction

The most common forms of neurodegenerative dementia include Alzheimer’s disease (AD), frontotemporal dementia (FTD), and dementia with Lewy bodies (DLB) [1,2,3]. These neurodegenerative disorders share similarities, such as aggregation of protein deposits in the affected brain and vulnerability of neurons in specific brain areas, and thus they are often referred to as proteinopathies. The differential diagnosis between these diseases is challenging due the overlapping clinical symptoms and the lack of specificity in the routinely used diagnostic tools. Especially, the discrimination of different types of early-onset dementias, which are not caused by specific autosomal dominant genetic mutations, has remained demanding. Making the correct diagnosis is essential as it helps both the healthcare personnel as well as the patients and their next of kin to deal with the patients’ symptoms. In addition, there is a great need to obtain the correct diagnosis already in the early phase of the disease.

The main strategy applied in dementia diagnostics over the years has been to assess the key protein pathologies specifically related to each type of neurodegenerative disorder. AD is characterized by extracellular deposits of β-amyloid (Aβ) and intracellular accumulation of phosphorylated Tau (pTau). The neuropathological inclusions in FTD typically contain tau or transactive response DNA-binding protein 43 (TDP-43) [4]. The neuropathological hallmarks of DLB include α-synuclein-positive Lewy bodies and neurites, and neuronal loss, but mixed pathology of DLB and AD is also frequently observed [5]. 

Enzyme-linked immunosorbent assay (ELISA)-based analyses of specific AD-associated proteins, namely cerebrospinal fluid (CSF) Aβ42 and tau protein, have been the first routinely used approaches in fluid biomarker studies in dementia as well as in clinical daily work [1]. However, the invasive nature of the lumbar puncture restricts the deployment and repetition of the CSF measurements. 

The research on diagnostic markers in dementia has made a giant leap forward in the recent years. The limited possibilities to detect small amounts of targets with traditional methods has recently prompted the development of various novel methods capable of detecting ultra-low quantities of specific biomarker targets from different biofluids [6]. Importantly, this has led to increased possibilities to indirectly analyze central nervous system (CNS)-derived biomarkers from blood in addition to CSF samples.

Currently, there are no drugs available to stop or decelerate the disease progression, and the management of different types of dementia varies. In the future, disease-modifying therapies, affecting the pathogenic mechanisms, will most likely be disease-specific and used in a personalized manner. Therefore, achieving the correct diagnosis early is important for designing drug trials. Moreover, the correct timing of the initiation of the future drug treatment is important in patients with autosomal dominant genetic forms of dementia. Specific and suitable biomarkers, especially indicating decreased neuronal injury, would be needed for the assessment of the effect of these drug treatments. An optimal fluid biomarker should be easily obtained (e.g., blood sample) and it should preferably also have prognostic value. 

In this review, we briefly analyze the traditional as well as the most potential and advanced recent techniques in the diagnostics of dementia developed over the years (see Figure 1 and Table 1), and we suggest an algorithm based on these state-of-the-art blood biomarkers to be considered for use in clinical research and hopefully in future clinical routine (see Figure 2).

## 2. ELISA-Based Detection of Markers in Biofluids

ELISA is thus far the only largely approved method utilized in the diagnostics of common neurodegenerative diseases. The method was initially introduced in 1970s [7] and is based on binding of highly specific antibodies to their respective antigens in the sample.

In the analysis, the target antigen in the sample is bound to a solid surface, such as a microplate, directly or via a capture antibody immobilized on the plate. The antigen is then detected directly using an antibody containing a label or indirectly with the help of a labeled secondary antibody binding to the primary detection antibody. The so called “sandwich ELISA”, where the target antigen is immobilized on the plate surface by one primary antibody and detected using another primary antibody binding to another epitope in the antigen, is often used because of its enhanced specificity and sensitivity. As labels bound to either the primary or the secondary antibodies, enzymes, such as horseradish peroxidase (HRP) or alkaline phosphatase (AP), or different fluorescent labels can be used. Finally, the plates are incubated with suitable substrates for the label enzymes, which produce a colorimetric reaction as the result of the enzymatic activity, and this is then read using a spectrophotometer. In the case of fluorescently labeled antibodies, the signal is detected by a fluorometer. In both cases, the intensity of the color or fluorescent signal is proportional to the concentration of the target in the sample [8]. 

Vandermeeren and colleagues introduced initial ELISA-based CSF tau measurements in the diagnostics of AD in 1993 [9], which normally stabilizes the structure of the axonal microtubules and when released to the extracellular space, reflects the formation of the neurofibrillary tangles, one of the main neuropathological hallmarks in AD. Subsequently, in 1995, the first report of ELISA-based CSF Aβ42 in AD was published, indicating that decreased CSF Aβ42 levels in patients with AD were linked to increased β-amyloid pathology in the brain [10], which is another neuropathological hallmark in AD. Since the initial reports, there are numerous studies confirming the utility of CSF Aβ42 and both total and phosphorylated Tau in the diagnostics of AD by providing high sensitivity and specificity, greater than 85%, when comparing AD patients and healthy controls [11]. 

However, increased CSF Tau levels are not a typical feature of AD patients only and they have been detected in other conditions including ischemic stroke [12], FTD [13] and Creutzfeldt-Jakob disease (CJD) [14,15]. This suggests that elevated CSF Tau levels associate with neuronal damage in general rather than with a specific neurological or neurodegenerative disorder. Moreover, in the carriers of the *C9orf72* repeat expansion, which is one of the most common genetic etiology of familial FTD, decreased CSF Aβ42 levels, showing a similar change to AD patients, have been detected [13]. These findings suggest that also CSF Aβ42 measurements may not only reflect AD-type pathological changes. 

Nonetheless, CSF Aβ42 and Tau measurements are currently considered a valid and accurate biomarker for the diagnosis of AD, along with amyloid Positron Emission Tomography (PET) tracers [1,16,17], but they display insufficient specificity between various neurological disorders and dementia [11,18,19,20]. In addition, the lack of prognostic value and the invasiveness of CSF AD biomarker measurements restrict the practicability of this method. Also, medications such as anticoagulative therapeutics may restrict the use of CSF procedures, especially since monitoring of the effects of eventual future disease-modifying drugs would need repeated sampling. Finally, because the CSF samples are not typically obtained in the basic health care, these measurements cannot be widely utilized in the screening of dementing disorders. In conclusion, these drawbacks have raised the need for more practical tools for the screening of possible neurodegenerative disorders in individuals with subjective cognitive complaints.

## 3. Single-Molecule Enzyme-Linked Immunosorbent Assay (SIMOA): The New Avenue to Peripheral Biomarkers in Neurodegenerative Dementias

SIMOA was initially presented in 2010 [21]. The SIMOA method is particularly effective and ultrasensitive in the detection of very small quantities of proteins, being able to detect even single molecules at the fg/mL range in biofluids and other samples.

The method is based on the detection of single enzyme-linked immunocomplexes on beads that are sealed in arrays of femtoliter-volume wells. Due to the extremely high sensitivity, the SIMOA method is useful to detect such proteins from CSF samples that are under the threshold of detection by traditional methods. In addition, the SIMOA method allows the detection of ultra-low quantities of brain-derived molecules from blood samples that are normally detectable only in CSF samples. The possibility to use blood samples has enabled the utilization of the SIMOA method for the discovery and identification of peripheral biomarkers and to test the accuracy of previously used CSF markers in the peripheral system, i.e., blood. Moreover, this approach will allow biomarker screening in a large scale also in the basic health care and, as a result, could possibly reduce the need to obtain CSF samples for research purposes.

In the analysis process, the measured proteins are first captured on microscopic beads coated with target-specific antibodies, forming single-molecule immunocomplexes. In very small concentrations, the beads may contain either one or none of the examined protein molecules (Poisson distribution) [22]. When each bead contains only a maximum of one molecule, these are not detectable by conventional enzyme labelling methods because the fluorescence signal is too low. In SIMOA, beads are loaded together with the fluorogenic enzyme substrate onto a disc array containing femtoliter-sized microwells, fitting only one bead (2 mm array containing approximately 50,000 wells). Beads holding an enzyme-labeled immunocomplex will produce a high local concentration of fluorescent product, restricted to reaction chamber. By utilizing time-lapsed fluorescence images with standard microscope optics, it is possible to discriminate the positive (on-well) beads from those not linked to the enzyme (off-well). The imaging method is capable to simultaneously detect the signal from tens to tens of thousands immunocomplexes. Finally, the concentration of the target protein is determined by counting the number of wells with a bead combined with fluorescent product and comparing the number to the total number of wells containing beads [21].

First, in the assessment of the utility of SIMOA in the diagnostics of neurodegenerative diseases, the method was shown to provide an accurate correlation between the CSF and plasma Aβ42, Aβ40, and their ratio [23,24]. However, extra-cerebral amyloid production, for instance from the platelets [25], has made this blood sample-based measurement a less promising tool in the AD diagnostics because the extra-cerebral production sources are difficult to control in the measurements. 

The present literature does not encourage using blood-based SIMOA Aβ measurements in the discrimination and diagnostics of neurodegenerative dementias. We found only one report specifically describing SIMOA-based plasma Aβ40 measurements in AD and control subjects, in which the researchers did not find relevant differences [26]. Furthermore, a recent meta-analysis by Koychev and colleagues [27] did not find any difference in plasma Aβ42 levels between AD patients and controls, based on two previously published reports describing SIMOA-based measurements [26,28]. Also, the Aβ42/40 ratio did not significantly differ between AD patients and healthy controls [26], but it might be able to discriminate those mild cognitive impairment (MCI) patients that convert to AD in the follow-up [29] and associate with positive amyloid-PET imaging in preclinical AD patients [30].

SIMOA-based Tau measurements have provided more promising data in the AD diagnostics. First reports indicated that SIMOA-based plasma Tau measurement was capable to separate groups of cognitively impaired and cognitively normal cases [31,32] as well as to associate with decreased cortical volume [31]. Surprisingly, Shi and colleagues reported that SIMOA-based Tau measurements could be elevated even more in patients with Parkinson’s disease than in AD patients [33]. However, these results were later questioned as no changes in the peripheral blood sample Tau levels were found between the patients with subjective cognitive decline and controls [34]. Later, various research groups concluded that plasma Tau does not have diagnostic potential in AD [26,28,35,36] nor in the FTD spectrum disorders [37]. It also has remarkably less diagnostic potential compared to plasma pTau181 [38]. In a meta-analysis, SIMOA-based total plasma Tau levels were shown to be elevated in patients with AD. If serum Tau is analyzed, the combination with conventional ELISA-based CSF measurements may improve the sensitivity of AD detection [39].

Even though total Tau measurements in the plasma did not suggest diagnostic potential, the levels of Tau phosphorylated at threonine 181 (plasma pTau181), a novel neuroaxonal damage biomarker, have been reported to successfully separate AD and FTD patients from each other as well as AD patients from healthy controls [38,40,41]. Plasma pTau181 has a high specificity for AD (compared to CSF Tau) and sensitivity [42,43,44,45], suggesting that it might possess potential in the differential diagnostics between AD and other types of dementia, especially compared to traditional CSF-based Tau measurements. In a substantially large study with 309 participants (MCI, AD, or healthy controls), plasma pTau181 was shown to have an AUC of 0.91 and higher levels associated with a more severe cognitive decline and gray matter loss in temporal regions [46]. The same group also reported elevated plasma pTau181 levels in MCI cases who later converted to AD, which indicates potential prognostic value. A recent meta-analysis concluded that plasma pTau181 may be useful in the diagnostics of dementia [27]. However, this report underlined that the optimal method to measure plasma pTau181 is thus far unclear, and methods other than SIMOA (including ELISA) may be utilized in the measurements. 

Along with pTau181, other novel Tau-based biomarkers measurable by SIMOA have been recently introduced for the enhanced diagnostics of AD. These include Tau phosphorylated at threonine 217 (pTau217) in CSF [47,48], plasma N-terminal fragment of Tau (NT1) [49], plasma p-Tau231 [48], and N-terminal Tau fragments ending at amino-acid 224 (N-224) in the CSF [50] and plasma [51]. However, these novel Tau markers need further validation as their discriminative accuracy remains thus far elusive.

At the moment, SIMOA approaches have been tried for other proteins involved in FTD. For example, companies such as Quanterix have recently developed SIMOA detection kits for TDP-43 based on a full-length protein calibrator and antibodies against AA 203–209 and the C-terminal region. However, the clinical efficacy of this approach still remains to be tested.

Another neurodegeneration-related protein for which a SIMOA approach has been recently developed is neurofilament light chain (NfL) protein. NfL is a protein that is part of the cellular cytoskeleton, providing structural support to axons and regulating axonal diameter [52]. NfL has been reported to be elevated in CSF and serum in various neurological conditions, indicating neuronal and axonal injury. Beyond papers describing CSF NfL as a reliable marker of neurodegeneration [53,54], serum NfL levels have been found elevated in AD [28,36,37,55,56,57,58,59,60], vascular dementia [61], Parkinson’s disease dementia [56] and FTD [37,40,62] when compared to healthy controls. Importantly in FTD, serum NfL levels have been shown to correlate with cognitive deficits, disease severity, behavioural symptoms, and the severity of frontotemporal atrophy [40]. Moreover, serum NfL levels correlate with the age of the patient and global cognitive disturbances [37,55,56,57,62,63,64] as well as with the severity of neurofibrillary tangle pathology in post-mortem brain [57].

Overall, the current literature supports the idea that NfL is a general marker of neurodegeneration and disease severity but may not be considered a disease-specific diagnostic marker. Indeed, the disadvantage of NfL as a biomarker is that its levels are elevated in various situations causing neuroaxonal damage, including for instance traumatic brain injury [65] and multiple sclerosis [66]. However, altered NfL levels have been shown to discriminate patients with FTD from those with primary psychiatric disorders [64]. The discrimination of these patients is often clinically challenging but very relevant regarding setting of the correct diagnosis and management of these patients. While plasma NfL levels also correlate with brain volume loss [37,67,68], blood-based NfL might prove as a practical tool in the monitoring of the effects of disease-modifying therapeutics in drug trials, even when no clear correlation between longitudinal cognitive decline and blood NfL levels has been shown [68]. However, in prodromal AD, SIMOA-based blood NfL levels have been shown to reflect longitudinal cognitive decline [58].

Furthermore, blood NfL may represent a practical tool for discriminating patients having subjective complaints without an underlying neurodegenerative etiology from those caused by neuroaxonal damage. The minimal invasiveness and the fact that samples can be obtained in conjunction with routine blood collection enables repeated sample analyses for monitoring the disease activity. 

Another interesting target using the SIMOA technology is represented by glial fibrillary acidic protein (GFAP). Recently, GFAP presence in serum, which reflects astrocytosis, has been shown to be elevated both in AD [69] and in FTD patients [70,71], as well as in patients with pre-symptomatic stages of AD [72]. Because serum GFAP levels have been shown to increase also in non-dementing conditions causing neuronal damage [73,74,75], they may eventually provide similar utility as NfL levels in the diagnostics of dementia, i.e., in reflecting the activity and prognosis of the neurodegenerative process. However, more research is needed to confirm the usefulness of GFAP measurements in biomarker studies.

## 4. Real-Time Quaking (RT-QuIC): A New and Promising State-of-the-Art Tool

RT-QuIC is a biochemical assay, originally developed for detecting antemortem prion pathology by measuring misfolded proteins (PrP) in Creutzfeldt-Jakob disease (CJD) [76,77]. The method is based on shaking of a small amount of test specimen with an excess of recombinant proteins. The misfolded proteins in the samples, containing β-sheet-rich structures, will further seed the assembly of the recombinant proteins and are bound by thioflavin T (ThT), which allows their fluorescent detection [78]. The method utilizes frequent “real-time” measurements of ThT fluorescence, allowing kinetic distinction between prion-seeded and unseeded fibrillization reactions as the unseeded reactions are markedly slower [78]. For example, 15 μL samples of CSF from sporadic CJD patients typically provide increased ThT fluorescence in less than 10 h, while control samples remain negative for at least 55 h [79]. 

As diagnostic tests, RT-QuIC assays have proved to possess extremely high specificity (100%) and substantially high sensitivity (95%) in sporadic CJD [80,81,82]. Importantly, the diagnostic sensitivity may be increased to 100% by combining different specimen types (e.g., CSF and nasal swab) from the patient [83,84]. 

In addition, also skin samples and olfactory mucosa swab have been proven to work as a substrate for RT-QuIC assay in sporadic CJD [85,86], and thus the possibility to use different types of biofluids and samples may represent an obvious advantage of this method. As a consequence of high diagnostic accuracy and utility, the amended diagnostic criteria for sporadic CJD acknowledged the utilization of RT-QuIC assay in the setting of diagnosis [87].

Since 2016, RT-QuIC has been shown to be a practical method also in detecting other misfolded proteins having seeding activity, namely CSF α-synuclein in neurodegenerative diseases with Lewy body pathology [88]. The initial report suggested 92% sensitivity for DLB and 95% sensitivity for Parkinson’s disease in the CSF α-synuclein measurements, and even 100% specificity when comparing these patients to AD patients. An improved assay in 2018 indicated even a higher accuracy, reaching 93% sensitivity in CSF measurements [89]. Since then, encouraging results in detecting CSF α-synuclein in the parkinsonism spectrum disorders have been reported [90,91]. Furthermore, it was demonstrated that similarly to sporadic CJD, the RT-QuIC method in human samples is not specimen type-specific as α-synuclein was detectable both in the skin tissue [92] and brain samples [93,94]. Later, a study by Rossi and colleagues [95] indicated that there were no differences in the seeding activity between different clinical syndromes, which are underlain by Lewy body pathology, suggesting that the method allows discrimination of different types of pathologies rather than clinical syndromes. 

Since 2017, an RT-QuIC method detecting different isoforms of the Tau protein has been available and was initially used to detect Pick’s disease-specific 3R-Tau isoforms [96]. As Tau aggregates are common in various neurodegenerative diseases (including FTD and AD), the presence of different Tau isoforms may specifically represent distinctive types of neurodegenerative diseases. For instance, 3R-Tau is detected in Pick’s disease, 4R-Tau in PSP and FTD, while AD and chronic traumatic encephalopathy represent mixed forms of 3R-Tau and 4R-Tau pathologies [97,98]. A high sensitivity of Tau detection in RT-QuIC-based method has been shown, but challenges in the detection of various isoforms have been pointed out [99,100]. However, an assay with a substantially enhanced sensitivity for specific detection of both 3R-Tau and 4R-Tau aggregates was recently introduced [101]. Moreover, an assay detecting all isoforms of Tau without discrimination of the different isoforms has been reported [102].

TDP-43 is a major neuropathological protein accumulating in familial FTD (linked to mutations in *GRN* and *C9orf72* genes) as well as in amyotrophic lateral sclerosis (ALS), and it normally contains various physiological functions such as RNA translation, autophagy, and synaptic plasticity [103]. Recently, also this protein was shown to be highly detectable in the CSF via an RT-QuIC method [104]. The study by Scialò et al. reports that RT-QuIC is the first diagnostic tool, which can be used to accurately detect antemortem neuropathological changes in TDP-43-associated brains. They reported 94% sensitivity and 85% specificity in the detection of TDP-43 in CSF. The method was able to detect as little as 15 pg of the TDP-43 protein in the samples. 

Taken together, methods based on the utilization of prion-like seeding properties of disease-specific misfolded proteins in common neurodegenerative diseases can provide new and promising state-of-the-art tools for enhanced diagnostic procedures in clinical use and trials. However, for wider and further development, there are also some drawbacks that will need to be addressed. For example, to date the prognostic utility of RT-QuIC is unclear. Also, most of the studies have been based on CSF samples, which limits obtaining repetitive samples from the same patients for follow-up studies.

## 5. Conclusions

The specific diagnostics in different neurodegenerative diseases has remained challenging. Definite diagnoses cannot be currently set without detection of explicit disease-causing gene mutations or neuropathological (post-mortem) analyses. Diagnostic protocols desperately need new tools due to the fact that recent findings have underlined overlapping clinical and neuropsychological presentations in patients with different neurodegenerative diseases [105,106,107]. Also neuroimaging proves to have limited sensitivity especially in the early stages of the disease [108,109].

Early and accurate diagnosis is pivotal for the disease management of neurodegenerative disorders, even when disease-modifying drugs are not yet available. Most importantly, correct diagnosis is essential in the clinical trials related to novel therapeutics modifying the disease course and when applying these medications in the future to the patients before irreversible neuronal damage and loss leading to cognitive deficits. 

The establishment of ELISA-based measurements of Tau and Aβ in AD in the mid 1990s provided new approaches for the diagnostics of neurodegenerative dementia [9,10] and the possibility to detect especially AD already in the MCI phase [110,111]. However, the major limitations of traditional ELISA-based CSF AD biomarker analyses include incomplete discrimination between different neurodegenerative dementias as well as the invasive nature of CSF sampling. Moreover, significant limitations have been found for important proteins in FTD like TDP-43, where trying to measure disease-specific forms of this protein in biofluids (plasma and CSF) using ELISA techniques has not allowed to obtain consistent results (as reviewed in [112]). Furthermore, the CSF analyses often lack prognostic value.

In the last few years, however, ground-breaking new approaches on biomarker-based diagnostics of neurodegenerative dementias have been introduced (see Figure 1). In this review, we have presented these novel molecules based on the most promising methods for diagnostics, i.e., SIMOA and RT-QuIC (see Table 1). 

We suggest that a systematic utilization of a diagnostic algorithm would enhance the accuracy of clinical trials and could be employed also in the routine clinical work. Therefore, we propose a two-step fluid biomarker-based algorithm (see Figure 2). First, the presence or absence of an ongoing neurodegenerative disorder should be proven. Blood NfL may be considered as a candidate biomarker at this stage. The second step is the differential diagnosis between neurodegenerative disorders, in which the best candidate biomarker should be chosen based on the clinical assessment.

The algorithm should be used in accordance with prevailing diagnostic pathways, including clinical and neuropsychological assessment and routine neuroimaging evaluation. We believe that the utilization of this biomarker algorithm would accelerate the diagnostics and research, and additionally save societal expenses by enabling earlier diagnoses [113]. In addition, especially clinical drug trials might benefit from the use of the presented algorithm. However, we acknowledge that this algorithm is at a research stage, and only a limited number of centers perform RT-QuIC or SIMOA analyses. In future, machine learning may provide a tool to include a larger number of biomarkers to the diagnostic assessment.

All in all, new biomarkers detected by novel and sensitive state-of-the-art methods hold great potential to change the current clinical practice. Future studies are needed to confirm the present data and allow the development of novel algorithms for the screening, diagnosis, and monitoring of neurodegenerative dementias.

## Figures and Tables

**Figure 1 diagnostics-11-00788-f001:**
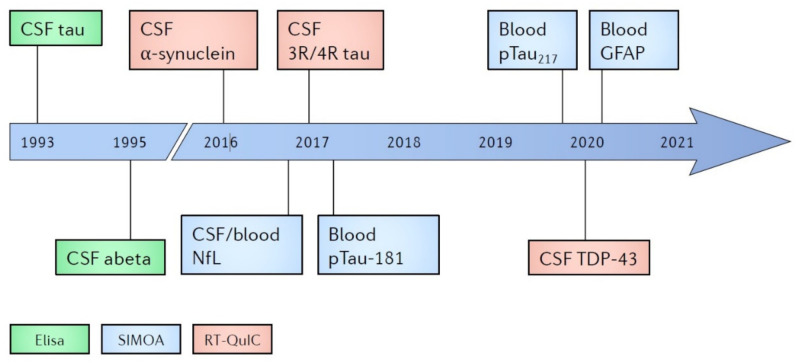
Timeline of the diagnostic biomarkers and methods. Aβ, beta-amyloid; CSF, cerebrospinal fluid; ELISA, enzyme-linked immunosorbent assay; GFAP, glial fibrillary acidic protein; NfL, neurofilament light chain; pTau181, Tau phosphorylated at threonine 181; pTau217, Tau phosphorylated at threonine 217; RT-QuIC, real-time quaking-induced conversion; SIMOA, single-molecule enzyme-linked immunosorbent assay; TDP-43, TAR DNA-binding protein 43.

**Figure 2 diagnostics-11-00788-f002:**
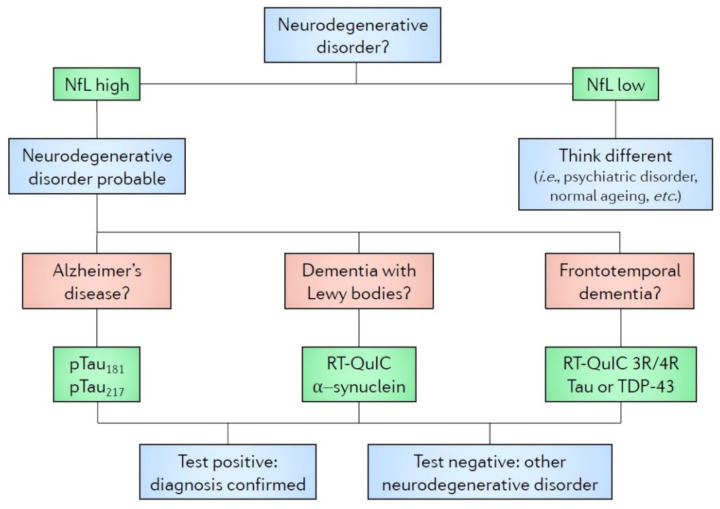
A proposed algorithm for diagnostics and prognostic assessment of neurodegenerative dementias. AD, Alzheimer’s disease; CSF, cerebrospinal fluid; DLB, dementia with Lewy bodies; FTD, frontotemporal dementia; NfL, neurofilament light chain; pTau181, tau phosphorylated at threonine 181; pTau217, tau phosphorylated at threonine 217; RT-QuIC, real-time quaking-induced conversion; TDP-43, TAR DNA-binding protein 43.

**Table 1 diagnostics-11-00788-t001:** The conventional and novel potential biomarkers for neurodegenerative dementia. Aβ42, beta-amyloid 42; AD, Alzheimer’s disease; CSF, cerebrospinal fluid; DLB, dementia with Lewy bodies; ELISA, enzyme-linked immunosorbent assay; FTD, frontotemporal dementia; GFAP, glial fibrillary acidic protein; NfL, neurofilament light chain; pTau181, Tau phosphorylated at threonine 181; pTau217, Tau phosphorylated at threonine 217; RT-QuIC, real-time quaking-induced conversion; SIMOA, single-molecule enzyme-linked immunosorbent assay; TDP-43, TAR DNA-binding protein 43.

Marker	Diagnostic	Neurodegeneration	Prognostic
***ELISA***			
CSF Aβ42 CSF tau	AD		
		Yes	
***SIMOA***			
CSF and blood NfL Blood GFAP Blood pTau_181_ Blood pTau_217_		Yes	AD, FTD
		Yes	
	AD		AD
	AD		
***RT-QuIC***			
CSF α-synuclein	DLB		
CSF 3R/4R tau	AD, FTD		
CSF TDP-43	FTD		

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
