# Peer review of "State-of-the-Art Methods and Emerging Fluid Biomarkers in the Diagnostics of Dementia—A Short Review and Diagnostic Algorithm"

_diagnostics, 2021, doi:10.3390/diagnostics11050788_

Round 1

Reviewer 1 Report

In this paper, dementia diseases were comprehensively overviewed in the course of biomarkers search and discovery.

SIMOA and RT-QuIC approaches were highlighted as modern analytical schemes in this field.

CSF and blood were described as samples-of-choice for the diagnostics of dementia based on biomarker discovery.

The topic is highly actual and diagnostics based on the biomarker search and discovery presents the present and the future approach-of-choice where the early diagnostics can be done by analysing molecular compounds unique for the particular disease.

Introduction includes general description of diagnosis of neurodegenerative dementia diseases. In subsequent subchapters application of ELISA, SIMOA and RT-QuIC approaches to search for biomarkers in neurodegenerative dementias are briefly described.

Even you declare, that the work is brief it needs more information to give about the disease and approaches.

For example, you should provide some statistics about the worldwide cases of dementias and what are the trends in the course of dementia based on their increase/decrease.

“Following the limited possibilities to detect small amounts of targets with traditional methods, current status has recently prompted the development of various novel methods capable of detecting ultra-low quantities of specific biomarker targets from different biofluids and tissues [https://doi.org/10.1002/pmic.202000318]. Importantly, increased possibilities to indirectly analyze central nervous system (CNS)-derived biomarkers from blood in addition to CSF samples has a growing trend [?].”

Among others, the above paragraph has no citation. Please, supply references to sections with citation “gap”. Also, the above section perfectly fits to recent approach based on “microproteomics” which uses a very small samples to biomarker discovery, thus I made an upgrade which I stress important.

Also I wonder about other sensitive approaches in the field. What about proteomics biomarker discovery research? I believe there are several valuable publications in the scope.

You should also give some information about biomarker basic characteristics from general perspectives.

If this is a short report on the overview of the methods, you should state this in the title: “State-of-the-art methods and emerging fluid biomarkers in the diagnostics of dementia – a mini-overview”

I value graphics which is simply made and thus easily comprehend to the reader. However, draw a schemes of the workflows of the described approaches as ELISA, SIMOA and RT-QuIC, this also helps reader to make a quick glance with quick understanding.

Make a short table of “pros-n-cons” of the described approaches. Such graphics is also helpful to give a quick opinion of the authors about the benefits and shortages regarding chosen topic. The review needs to contain such opinion of authors about the status of the approaches as it is basically their opinion to be published.

Author Response

Comment 1:

In this paper, dementia diseases were comprehensively overviewed in the course of biomarkers search and discovery.

SIMOA and RT-QuIC approaches were highlighted as modern analytical schemes in this field.

CSF and blood were described as samples-of-choice for the diagnostics of dementia based on biomarker discovery.

The topic is highly actual and diagnostics based on the biomarker search and discovery presents the present and the future approach-of-choice where the early diagnostics can be done by analysing molecular compounds unique for the particular disease.

Introduction includes general description of diagnosis of neurodegenerative dementia diseases. In subsequent subchapters application of ELISA, SIMOA and RT-QuIC approaches to search for biomarkers in neurodegenerative dementias are briefly described.

Response: We thank the Reviewer for these comments.

Comment 2:

Even you declare, that the work is brief it needs more information to give about the disease and approaches.

For example, you should provide some statistics about the worldwide cases of dementias and what are the trends in the course of dementia based on their increase/decrease.

Response: We thank the Reviewer for rising up this point. Since this review focuses on subtypes of dementia, there is no comprehensive recent data on the subtypes. However, we feel optimistic that in close future the epidemiological data on subtypes will increase remarkably. In addition, we have reframed the title to describe better the nature of this review (see response to the comment 6).

Comment 3:

“Following the limited possibilities to detect small amounts of targets with traditional methods, current status has recently prompted the development of various novel methods capable of detecting ultra-low quantities of specific biomarker targets from different biofluids and tissues [https://doi.org/10.1002/pmic.202000318]. Importantly, increased possibilities to indirectly analyze central nervous system (CNS)-derived biomarkers from blood in addition to CSF samples has a growing trend [?].”

Among others, the above paragraph has no citation. Please, supply references to sections with citation “gap”. Also, the above section perfectly fits to recent approach based on “microproteomics” which uses a very small samples to biomarker discovery, thus I made an upgrade which I stress important.

Response: We thank the Reviewer for this comment. This citation is now added to the manuscript (p.2, r.63)

Comment 4:

Also I wonder about other sensitive approaches in the field. What about proteomics biomarker discovery research? I believe there are several valuable publications in the scope.

Response: We agree that there are several potential approaches in the field of biomarkers, that could aid the diagnostics of neurodegenerative diseases. However, in this review we aimed to include only the methods that are already now translatable to the clinical work and/or clinical trials.  We have now edited the latest sentence of the introduction to highlight this (p.2, r.76-77.).

Comment 5:

You should also give some information about biomarker basic characteristics from general perspectives.

Response: We thank the Reviewer for this note. We have added information about the discussed biomarker to the manuscript (p.3, r. 106-107; p.4, r.111-112; p.6, r.229-230; p.7, r.316-318)

Comment 6:

If this is a short report on the overview of the methods, you should state this in the title: “State-of-the-art methods and emerging fluid biomarkers in the diagnostics of dementia – a mini-overview”

Response: We agree that the title could be reframed. The new form of the title: “State-of-the-art methods and emerging fluid biomarkers in the diagnostics of dementia – a short review and diagnostic algorithm”

Comment 7:

I value graphics which is simply made and thus easily comprehend to the reader. However, draw a schemes of the workflows of the described approaches as ELISA, SIMOA and RT-QuIC, this also helps reader to make a quick glance with quick understanding.

Response: We thank the Reviewer for this interesting suggestion. Since the topic of the review is the emerging biomarkers in the diagnostic setting, we believe that the technical images of the methods would perhaps turn the review slightly exhaustive for the readers. Also, the technical images would probably be out of the topic as the reviewer has been written especially for the clinicians and clinical researchers.

Comment 8:

Make a short table of “pros-n-cons” of the described approaches. Such graphics is also helpful to give a quick opinion of the authors about the benefits and shortages regarding chosen topic. The review needs to contain such opinion of authors about the status of the approaches as it is basically their opinion to be published.

Response: We thank the Reviewer of this comment. We agree that at some extent the text always reflects the opinion of the authors. However, we have reviewed all the published articles of the chosen biomarkers within the described method. Since the literature of these promising novel biomarkers is still somewhat limited, we don’t feel comfortable to judge pros-n-coins for the selected biomarkers or approaches. To date, only clear benefits and disadvantages of the described methods are technical and linked also to availability of maintenance ecc. which we feel to be quite out of the topic. No changes in the manuscript.

Reviewer 2 Report

The authors describe a comprehensive review on the latest findings of emerging fluid biomarkers for dementia. This is well summarized and organized. The readers will be very interested in this content.  If possible, it would be great to touch on the techniques using machine learning. 

Author Response

The authors describe a comprehensive review on the latest findings of emerging fluid biomarkers for dementia. This is well summarized and organized. The readers will be very interested in this content.  If possible, it would be great to touch on the techniques using machine learning.

Response: We thank the Reviewer for this encouraging comment. We agree that machine learning has a great potential in future to aid the diagnostics of neurodegenerative diseases. We have added a sentence “In future, machine learning may provide a tool to include a larger number of biomarkers to the diagnostic assessment” to the discussion -section (p. 10, r. 384-386).

Round 2

Reviewer 1 Report

The authors have accomplihed my remarks.

However, between ref 6 and 7 there is one additional reference to be given a number. I assume to correct this in the proof processing.

Otherwise it it ok